# Effect of Photo Irradiation on the Anaerobic Digestion of Waste Sewage Sludge-Reduced Methane and Hydrogen Sulfide Productions

**Shotaro Toya [1], Shunsuke Iriguchi [1], Kohei Yamaguchi [1] and Toshinari Maeda [1,2,*]**

[1] Department of Biological Functions Engineering, Graduate School of Life Science and Systems Engineering, 2-4 Hibikino, Wakamatsu, Kitakyushu 808-0196, Japan; p238042s@mail.kyutech.jp (S.T.); iriguchi.shunsuke488@mail.kyutech.jp (S.I.); yamaguchi.kohei567@mail.kyutech.jp (K.Y.)

[2] Collaborative Research Centre for Green Materials on Environmental Technology, Kyushu Institute of Technology, 2-4 Hibikino, Wakamatsu, Kitakyushu 808-0196, Japan

* Correspondence: toshi.maeda@life.kyutech.ac.jp; Tel.: +81-93-695-6064; Fax: +81-93-695-6008

**Abstract:** Since a large amount of sewage sludge (WSS) is generated daily, exploring effective methods for utilizing WSS is necessary. Although a photo-fermentation system sometimes alters the characteristics of microbial functions, there have been no attempts to perform photo-fermentation using WSS, which is regularly treated via dark fermentation. In this study, the effect of photo-fermentation (photo-irradiation) on anaerobic digestion using WSS was revealed. Photo-irradiation during the anaerobic digestion of WSS significantly reduced the amount of methane and hydrogen sulfide. Methane production was also reduced 5.6-fold at 13 days under light conditions, whereas hydrogen sulfide was consumed almost completely at 6 days. However, it was shown that the activity of sulfate-reducing bacteria in WSS under light treatment increased. Photo-irradiation also stimulated the growth of green sulfur bacteria and induced anoxygenic photosynthesis, via which process the fermented samples turned green in a manner that was correlated with their consumption of hydrogen sulfide. The production of organic acids was lowered in the samples that were irradiated using light. Finally, dark/light switching fermentation was only able to reduce hydrogen sulfide while methane production remained the same. The amounts of methane and hydrogen sulfide were 35 mmol/g VS, and they were undetected at 58 days in photo-irradiated samples compared to the control samples that produced 37 mmol/g VS of methane and 15 ppm/g VS of hydrogen sulfide.

**Keywords:** anaerobic digestion; photo-fermentation; dark fermentation; photo-irradiation; methane production; hydrogen sulfide consumption; waste sewage sludge

## 1. Introduction

Activated sludge systems, which enable the degradation of organic matter via the actions of aerobic bacteria, are used worldwide as a general wastewater treatment method [1,2]. However, a large amount of waste sewage sludge (WSS) is produced daily through these activated sludge systems [3]. In addition, WSS should be used effectively as a potential resource for a sustainable social system because a stable, large amount of WSS is generated on a daily basis [4]. Therefore, the development of a facile technology to effectively reduce the amount of WSS is an urgent issue. The anaerobic digestion process is known as an efficient reuse method of WSS because it can produce methane gas as a bioenergy resource [5]. However, the anaerobic digestion process is mostly operated in dark conditions, and there is a limited number of studies regarding anaerobic digestion processes illuminated by light. This is because the type of biogas to be targeted and the type of biomass resources to be used are different for each of the photo- and dark-fermentation systems [6]. In other words, the target of gas in the photo-fermentation system is biohydrogen instead of methane, and the biomass resources used are lignocellulosic biomasses, such as food waste [7].

Photo-fermentation systems can obtain biohydrogen from lignocellulosic biomasses [8]. The presence of sugars is important for biohydrogen production [9], and lignocellulosic biomass (such as corn, sugarcane bagasse, and wheat straw) is characterized by its high carbohydrate content through its cellulose and hemicellulose [10]. In addition, the aim of the photo-fermentation system is the growth of photosynthetic bacteria, which produce hydrogen from volatile organic acids [11]. Furthermore, it has been reported that light energy increases the catalytic reaction efficiency of nitrogenase in photosynthetic bacteria, which, in turn, promotes hydrogen production [12]. WSS is also a biomass resource consisting of polymeric compounds containing proteins, lipids, and carbohydrates [13]. As most WSS may include dead microorganisms, it contains a large amount of protein [14]. The anaerobic digestion process using WSS mainly consists of the following three stages: (a) the hydrolysis stage, which degrades the high-molecular-weight compounds (such as protein, carbohydrate, and lipid), which are components of WSS, to low-molecular-weight compounds, such as amino acids, sugars, and fatty acids; (b) the acidogenesis stage, which produces organic acids, such as acetic acid, from low-molecular-weight compounds; and (c) the methanogenesis stage, which produces methane from organic acids and other substances, which proceeds even in the absence of light energy [15].

For the improvement in the anaerobic digestion of WSS, various additives and approaches have been investigated, and detailed mechanisms have been clarified [16–18]. The antibiotic azithromycin has been shown to positively affect the acidogenic process and promote methanogenesis while chloramphenicol has been shown to decrease archaeal activity and inhibit methanogenesis [19], and the inorganic materials sodium tungstate and metallic material aluminum oxide have been shown to increase methane production by promoting the activity of acetoclastic methanogens and the hydrolysis stage, respectively [20,21]. In a study focusing on quorum sensing (QS) as a bacterial interaction, 5-fluorouracil as a QS inhibitor was observed to reduce the activity of acetoclastic methanogens, while the quorum-quenching enzyme AiiM increased the percentage of Gram-positive bacteria in the WSS and caused a decrease in methanogenesis [22,23]. In addition, it has been reported that pretreatments, such as thermal alkali, ultrasound, and decomposition enzymes for WSS, result in improved methane production [24–26]. Therefore, understanding the microbial interactions within the anaerobic digestion process using WSS has the potential to not only improve the anaerobic digestion process but also support the production of other useful substances (such as acetate, lactic acid, and bioethanol) from WSS and find new technological offerings.

The goal of this study is to investigate the effect of photo-fermentation (photo-irradiation) on the anaerobic digestion process using WSS in terms of microbial interactions and investigate the effect of photo-irradiation on the removal of hydrogen sulfide in anaerobic digestion. Methane production was suppressed, and the amount of hydrogen sulfide in the sample was significantly reduced via photo-irradiation during the anaerobic digestion. We report the effect of photo-irradiation on microbial interactions within anaerobic digestion systems. In this report, we summarize the interactions and relationships in the following four microbial groups: (1) acetoclastic methanogens, (2) sulfur-reducing bacteria, (3) green sulfur bacteria, and (4) organic-acid-producing bacteria. In addition, we report a dark/light switching fermentation that maintains methane but reduces hydrogen sulfide.

## 2. Materials and Methods

### 2.1. Waste Sewage Sludge (WSS) Preparation

WSS was kindly provided by the Hiagari wastewater treatment plant in Kitakyushu City, Japan. The WSS composition was reported to comprise the following elements: 3.3–4.0% total solid (TS), 2.6–3.1% volatile solid (VS), 30–41% suspended solid (SS), 26–33% volatile suspended solid (VSS), and 44–51 g/L chemical oxygen demand (COD). The main components of WSS are proteins (40–45%), carbohydrates (12–14%), and lipids (11–13% of total solids). Fresh WSS was washed three times via centrifugation at $8000 \times g$ for 10 min at 4 °C, and the pellet was resuspended in purified water prior to adjust its concentration to

50% (*w/w*) relative to water. Then, the WSS was finally adjusted to a concentration of 5% (*w/w*) in purified water.

### 2.2. Methane and Hydrogen Sulfide Production via Photo-Irradiation

The 5% WSS (30 mL) was prepared in 66 mL vials, and the vials were tightly sealed with rubber stoppers and crimped with aluminum caps. Then, nitrogen gas was sparged at 0.02 MPa for 15 s to create an anaerobic condition, and the vials were incubated at 37 °C for 13 days at 120 rpm with photo-irradiation at 2000–4000 lux illuminance using white LED (light-emitting diode) lamps (OHM Electric Inc., LDA7N g AG5, Yoshikawa-shi, Saitama, Japan). The vials were completely covered with aluminum foil and prepared as controls. For dark/light fermentation experiments, the vials were originally incubated for 52 days under dark conditions, and 0.5 mL of WSS with green sulfur bacteria (WSS, which was incubated for 7 days under the photo-irradiation) was added to the WSS sample. The mixture was additionally incubated for 6 days under light conditions. Each experiment was conducted in triplicate at minimum.

The amount of methane gas was measured by injecting 50 μL of headspace gas from each vial into a GC-3200 gas chromatograph (GL Science, Tokyo, Japan), as previously described in [19]. In addition, the amount of hydrogen sulfide in the headspace of each vial was measured using the GASTEC system (GASTEC Inc., GV-110S, Ayase, Japan), as previously described in [27]. The amount of each gas was calculated based on the volatile solid (VS) of WSS.

### 2.3. Chlorophyll Measurement

The amount of chlorophyll was measured to evaluate the growth level of photosynthetic bacteria using the method described previously in [28]. WSS samples with or without photo-irradiation (1 mL) were centrifuged at 13,000 rpm for 1 min, and the pellets and 100% methanol were mixed and incubated at room temperature for 5 min. After centrifugation at 10,000 rpm for 10 min, absorbance was detected at 650 nm and 655 nm using the supernatants.

### 2.4. Analytical Methods

WSS samples with or without photo-irradiation were centrifuged at 13,000 rpm for 1 min, and the supernatants that were filtered via a 0.2 μm membrane syringe filter were used for measuring protease activities and organic acids.

Protease activity was measured using casein as a substrate, as described previously in [19]. A casein solution (4%, 4 g in 100 mL of 0.4 M Tris-HCl, pH 8.5) was prepared. The casein (0.1 mL) and enzyme (0.3 mL) solutions were mixed and incubated at 37 °C for 120 min. After incubation, 0.44 M trichloroacetic acid (0.4 mL) was added to stop the reaction, and the solution was incubated at room temperature for 30 min. After centrifugation at 14,000 rpm for 10 min at 4 °C, the supernatant (0.2 mL), Folin reagent (0.2 mL), and 0.4 M sodium carbonate (1 mL) were mixed and incubated at 37 °C for 30 min. After the incubation, the absorbance was detected at 660 nm. One unit of protease activity was defined as the amount of tyrosine (μmol) produced from casein per minute using 1 mg of an enzyme.

Organic acids (acetate, propionate, isobutyrate, and butyrate) were analyzed using high-performance liquid chromatography (Shimadzu SCL-10ADvp, Kyoto, Japan), as described previously in [27].

### 2.5. Activity Measurement of Methanogens and Sulfur-Reducing Bacteria

The activities of acetoclastic methanogen and sulfur-reducing bacteria were analyzed to evaluate the effects of photo-irradiation on the microorganisms that directly produce methane gas and hydrogen sulfide.

For the assay of acetoclastic methanogen, WSS samples with or without photo-irradiation were incubated at 37 °C for 7 days at 120 rpm, and each sample was mixed

with four antibiotics (to the following final concentration: vancomycin at 0.2 mg/mL, ampicillin at 0.2 mg/mL, streptomycin at 0.5 mg/mL, and benzylpenicillin at 0.5 mg/mL) to inactivate bacterial activity [29]. Then, sodium acetate was added as a substrate to acetoclastic methanogen at a final concentration of 20 mM, and it was tightly sealed with rubber stoppers, crimped with aluminum caps, and sparged with nitrogen gas. The vials were additionally incubated at 37 °C for 7 days at 120 rpm under dark conditions, and the amount of methane gas was measured.

For the assay of hydrogenotrophic methanogen, the same WSS samples as those above were placed in vials, and the vials were tightly sealed with rubber stoppers, crimped with aluminum caps, and sparged with hydrogen/carbon dioxide gas (80%/20%). The vials were additionally incubated at 37 °C for 7 days at 120 rpm under dark conditions, and the amount of methane gas was measured.

For the assay of sulfur-reducing bacteria, WSS samples incubated for 7 days with or without photo-irradiation were washed three times using purified water, and $8000 \times g$ centrifugation for 10 min at 4 °C was carried out to remove soluble products (e.g., organic acids and soluble proteins). The pellets were resuspended in a substrate solution at a final WSS concentration of 5% ($w/w$); they were tightly sealed with rubber stoppers, crimped with aluminum caps, and sparged with nitrogen gas. The substrate solution contained sodium formate and sodium sulfate, each with a final concentration of 0.016 mM in the 5% WSS. The WSS sample adjusted to 5% using purified water was prepared as a blank sample. The vials were additionally incubated at 37 °C for 24 h at 120 rpm under dark conditions, and the amount of hydrogen sulfide was measured.

### 2.6. DNA Extractions

DNA was extracted from the pellets of WSS samples with or without photo-irradiation using a DNeasy PowerSoil Kit (Qiagen, Hilden, Germany) following the manufacturer's protocols. The extracted DNA samples were stored at −70 °C, and each piece of DNA was used as a template to investigate bacterial communities via MiSeq.

### 2.7. RNA Extraction and cDNA Synthesis

Total RNA was extracted from the pellets of WSS samples with or without photo-irradiation using the RNeasy kit (Qiagen, Valencia, CA, USA), as described previously in [27], and complementary deoxyribonucleic acid (cDNA) was synthesized using Prime-Script RT Reagent Kits (TAKARA Bio Inc., Shiga, Japan), as described previously in [19]. cDNA was used as a template to quantitate the archaeal population via quantitative real-time polymerase chain reaction (qRT-PCR).

### 2.8. qRT-PCR and High-Throughput 16S rRNA Sequencing and Data Processing

The qRT-PCR quantification for archaea in the samples was executed using the StepOne Real-Time PCR System (Applied Biosystem, Waltham, MA, USA) in order to amplify and detect fluorescence via the specific primers and probes of the TaqMan system. The real-time PCR mixture and cycling conditions were set as described previously in [19]. 16S rRNA genes were amplified using forward primer 341F (5′-CCTACGGGNGGCWGCAG-3′) and reverse primer 785R (5′-GACTACHVGGGTATCTAATCC-3′), targeting the V3 and V4 regions [30]. All steps for sequencing followed the Illumina protocol for the 16S ribosomal RNA gene sequencing library for the Illumina MiSeq system, as previously described [19]. The data obtained were de-multiplexed, and the reads were then classified into different taxonomic levels. Raw sequence data were deposited in the National Center for Biotechnology International (NCBI) Sequence Reads Archive (SRA) database under accession numbers SRR26189151 and SRR26189152. The obtained data were processed as previously described in [19].

## 3. Results and Discussion

### 3.1. Effect of Photo-Irradiation on Methane and Hydrogen Sulfide Productions

Firstly, because the anaerobic digestion of WSS produces methane as a main product and hydrogen sulfide as a by-product, the effect of photo-irradiation during the anaerobic digestion of WSS was investigated by focusing on the amounts of methane and hydrogen sulfide produced. As a result, methane production was suppressed, and the amount of hydrogen sulfide in the vial that was significantly reduced is photo-irradiation is shown in Figure 1. Methane production was observed after 4 days of incubation, whereas photo-irradiation suppressed methane production, and the amount of methane via photo-irradiation and the control sample during 13 days of incubation was $1.29 \pm 0.03$ vs. $7.2 \pm 0.5$ µmol/g VS, which represents a 5.6-fold difference (Figure 1a). In addition, the production of hydrogen sulfide was observed after 3 days of incubation in the samples with or without photo-irradiation: the amount of hydrogen sulfide via photo-irradiation and the control sample was $14.8 \pm 0.3$ vs. $13.3 \pm 0.1$ ppm/g VS, whereas the amount of hydrogen sulfide with photo-irradiation was almost undetectable after 6 days of incubation and recorded as $0.06 \pm 0.02$ ppm/g VS (Figure 1b). These results indicate that photo-irradiation has different effects on methane and hydrogen sulfide production during anaerobic digestion. In other words, methane production is inhibited from the beginning period via photo-irradiation, whereas a decrease in hydrogen sulfide occurs during incubation.

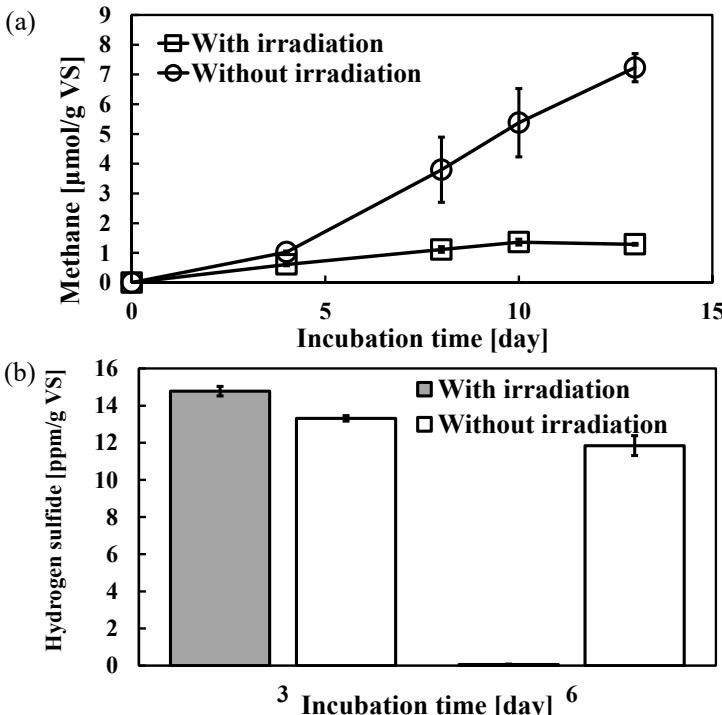

**Figure 1.** Effect of photo-irradiation on the anaerobic digestion of WAS: the amounts of methane (**a**) and hydrogen sulfide (**b**) were measured during incubation. Error bars represent standard errors (*n* = 3).

The change in the appearance of the photo-irradiation sample was remarkable, and no change was observed after 3 days of incubation, whereas discoloration to green was observed after 6 days of incubation due to the effect of photo-irradiation. This discoloration was caused by the growth of green sulfur bacteria, and the amount of chlorophyll in the photo-irradiation sample was significantly high (Figure 2). In addition, the bacterial community structures with photo-irradiation and the control sample after 13 days of incubation were analyzed, and it was indicated that *Chlorobiaceae*, which includes green sulfur bacterium, was the most abundant bacterial family in the photo-irradiation sample (Figure 3). Firstly, the data indicate that the bacterial community's structure in the WSS

sample with photo-irradiation is completely different compared to the control WSS. Green sulfur bacteria are known as photosynthetic sulfur bacteria, similarly to purple sulfur bacteria, and these bacterial communities are capable of anoxygenic photosynthesis [31]. Various sulfur compounds, including hydrogen sulfide, are used as electron donors during anoxygenic photosynthesis by photosynthetic sulfur bacteria [32]. Hence, as shown in Figure 1b, the reason why no decrease in hydrogen sulfide was observed during the 3 days of incubation may be that green sulfur bacteria do not grow during this period, and it was considered that anoxygenic photosynthesis by green sulfur bacteria occurred during the 6 days of incubation in which a discoloration to green was observed; moreover, hydrogen sulfide was consumed in the form of electron donors following incubation. Although *Candidatus Moranbacteria* was also detected as a bacterial family at approximately the same level as *Chlorobiaceae* in Figure 3, the metabolic functions of the organism under photo-irradiation are unknown. However, *Candidatus Moranbacteria* may contribute to the production of malate from carbon dioxide [33] because there are extra amounts of carbon dioxide due to the inhibition of methane production.

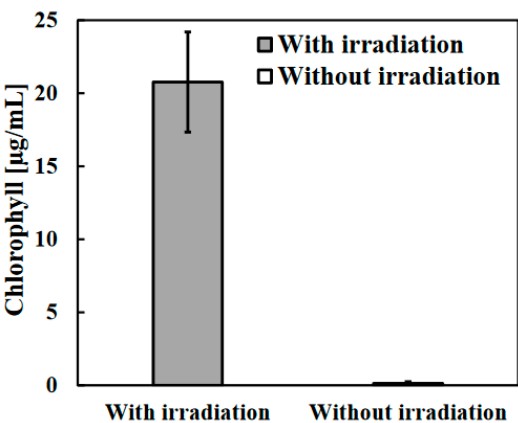

**Figure 2.** The amount of chlorophyll after 13-day anaerobic digestion in the samples of WSS with or without photo-irradiation. Error bars represent standard errors (*n* = 3).

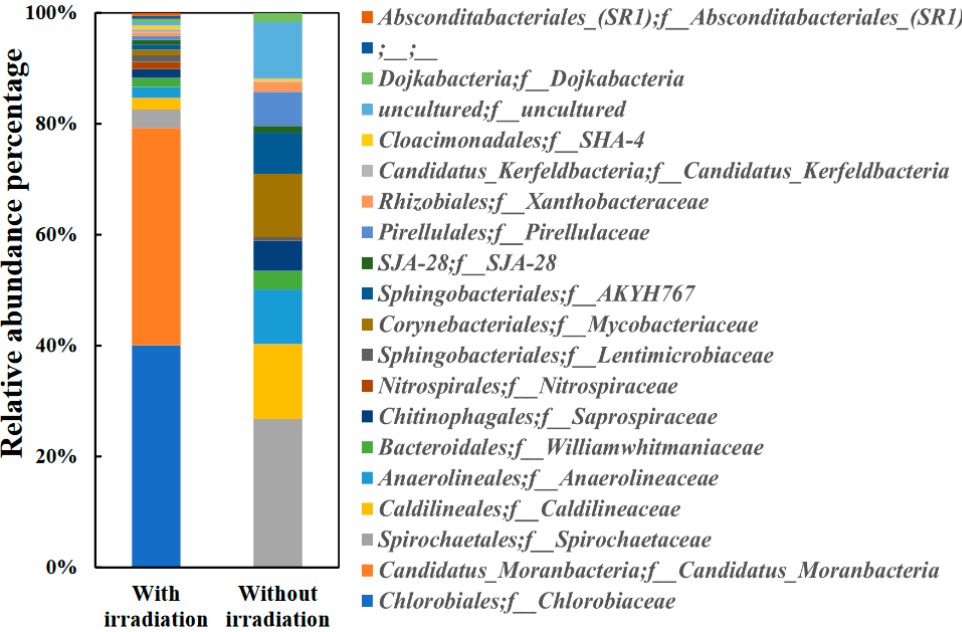

**Figure 3.** Dynamics of the bacterial community at 13 days of anaerobic digestion in WSS samples with or without photo-irradiation. Relative abundant percentages of the dominant bacterial community indicated by the order or family level.

Thus, photo-irradiation suppressed methane production during anaerobic digestion, while it accelerated the growth of green sulfur bacteria, which caused the consumption of hydrogen sulfide via anoxygenic photosynthesis.

### 3.2. Effect of Photo-Irradiation on Anaerobic Digestion

It was described earlier that methane production in 5% WSS was reduced via photo-irradiation (Figure 1a). The anaerobic digestion process using WSS mainly consists of three stages: the (a) hydrolysis stage, (b) acidogenesis stage, and (c) methanogenesis stage [13]. To clarify the suppression mechanism of methane production via photo-irradiation, the hydrolysis, acidogenesis, and methanogenesis stages during methane production using WSS were evaluated. Since proteins are the main components of WSS [12], protease activity was first measured to evaluate the hydrolysis stage.

Protease activity was slightly high in the photo-irradiation sample during the 3 days of incubation, whereas no significant difference in protease activity was observed between the photo-irradiation sample and the control sample during the 13 days of incubation (Figure 4a). Bacteria in sludge often use degrading enzymes (protease, amylase, cellulase, and lipase) to degrade the high-molecular-weight compounds in the WSS for their growth [34,35]. However, the presence of alternative nutrients may suppress the expression of degrading enzymes [36]. *Chlorobiaceae* grown via photo-irradiation are green sulfur bacteria (Figure 3), and their main growth strategy is anoxygenic photosynthesis using sulfur compounds [31]. In addition, there is hardly any knowledge of the production of degradative enzymes by green sulfur bacteria. Therefore, it was indicated that the hydrolysis stage during methane production was not affected by photo-irradiation.

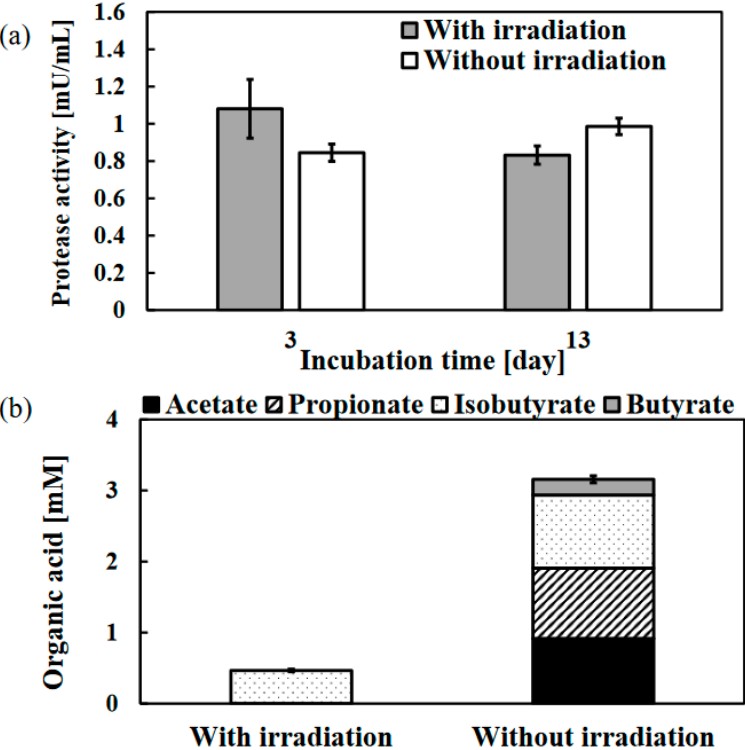

**Figure 4.** Effect of photo-irradiation on the hydrolysis (**a**) and acidogenesis stages (**b**) during the anaerobic digestion of WSS. Protease activity was measured at 3 and 13 days of incubation. These organic acids were measured at 13 days of incubation. Error bars represent standard errors (*n* = 3).

The measurement of organic acids produced during anaerobic digestion is a good target for evaluating the acidogenesis stage. Figure 4b shows the amount of organic acids in each sample after 13 days of incubation with or without photo-irradiation. The main organic acids detected were acetate, propionate, isobutyrate, and butyrate in the HPLC system,

and the results indicate that photo-irradiation significantly reduced the accumulation of organic acids (Figure 4b). Acetate, propionate, and butyrate accumulated in the control sample, while only isobutyrate was detected in the photo-irradiation sample. The presence of acetate is important for methanogenesis from organic acids, and methanogens, such as *Methanosarcina* and *Methanosaeta*, produce methane using acetate as a substrate [37]. In addition, propionate and isobutyrate have been reported to be converted to acetate by bacteria such as *Syntrophobacter* and *Smithella* [38,39], and the presence of acetate is one of the key points in methane production [40]. Since methane production was suppressed via photo-irradiation (Figure 1a), it is clear that the organic acids in the photo-irradiation sample were not used for methane production. In addition, it was reported that organic acids were utilized as nutrients by red non-sulfur bacteria, which are included in photosynthetic sulfur bacteria [41]. However, since the growth of green sulfur bacteria was promoted via photo-irradiation in this experimental system (Figure 3), it is unlikely that organic acids were consumed by the sulfur bacteria. Some reports state that acidogenesis may be affected by photo irradiation [42,43]. Therefore, although the detailed effects on organic acid-producing bacteria are unknown, it was indicated that photo-irradiation had a negative impact on the acidogenesis stage and suppressed organic acid accumulation. In addition, the archaeal activity in the samples during 13 days of incubation with or without photo-irradiation was quantified using qRT-PCR, and it was found that the archaeal activity of the photo-irradiation sample was 4.8 times lower than that of the control sample.

Thus, photo-irradiation during anaerobic digestion caused the suppression of organic acid production and a decrease in archaeal activity, thereby suppressing methane production.

### 3.3. Effect of Photo-Irradiation on Acetoclastic Methanogens and Sulfate-Reducing Bacteria

The amount of methane and hydrogen sulfide present during anaerobic digestion was markedly changed via photo-irradiation, and the growth of green sulfur bacteria and negative effects on the acidogenesis stage were involved. For example, although acetate is a substrate for the production of methane (acetoclastic methanogenesis) among organic acids, there is no detectable acetate in the sample with photo-irradiation (Figure 4). Two possibilities explain the reason why a lower amount of methane was produced: (1) Acetoclastic methanogens may be active; however, less methane production can be observed due to the inability to produce acetate. (2) Acetoclastic methanogens may be inactive, thereby lowering methane production.

In the case of the production of hydrogen sulfide, whether sulfur-reducing bacteria may be inactivated under light irradiation should be clarified. Figure 1b clearly shows that the amount of hydrogen sulfide is not detectable. During the process, there are at least two reactions, and the reactions should be competitive: (1) the production of hydrogen sulfide by sulfur-reducing bacteria and (2) the consumption of hydrogen sulfide by green sulfur bacteria. Therefore, in order to further clarify the effects of photo-irradiation on anaerobic digestion, the activities of acetoclastic methanogens and sulfate-reducing bacteria were evaluated. Each sample with or without photo-irradiation was used as a bacterial source; hence, the source was added to the assay buffers in which substrates for methane and hydrogen sulfide were present. These assays only allowed us to evaluate the presence of acetoclastic methanogens and sulfur-reducing bacteria in the bacterial source. The respective activities of acetoclastic methanogens and sulfate-reducing bacteria were measured as the amount of conversion to metabolic products (methane for acetoclastic methanogens and hydrogen sulfide for sulfate-reducing bacteria) under the presence of equal amounts of each substrate.

Figure 5a shows the acetoclastic methanogen activity in each sample after 7 days of incubation with or without photo-irradiation. The results showed that the amount of methane with respect to the photo-irradiation sample and the control sample was $3.4 \pm 0.8$ vs. $13 \pm 3$ μmol/g VS, and the methane production of the photo-irradiation sample was 3.9 times lower than that of the control sample despite an equal amount of

substrates (Figure 5a). No differences between the photo-irradiation sample and the control sample were observed during hydrogenotrophic methanogenesis, which produces methane using hydrogen as a substrate. The low concentration of acetate in the photo-irradiation sample (Figure 4b) may also cause reduced acetoclastic methanogen activity. Hence, it was confirmed that photo-irradiation particularly reduced acetoclastic methanogen activity. In addition, Figure 5b shows sulfate-reducing bacterial activity in each sample after 7 days of incubation with or without photo-irradiation. Interestingly, the hydrogen sulfide production of the photo-irradiation sample from an equal amount of substrate was 6.5 times higher than that of the control sample (Figure 5b). This result indicated that despite the markedly reduced amount of hydrogen sulfide present in the photo-irradiation sample (Figure 1b), sulfate-reducing bacteria were not affected by photo-irradiation. This result is consistent with a previous report that sulfate-reducing bacteria are not killed by only the light irradiation [44]. From these results, the effect of photo-irradiation on anaerobic digestion is summarized in Figure 6.

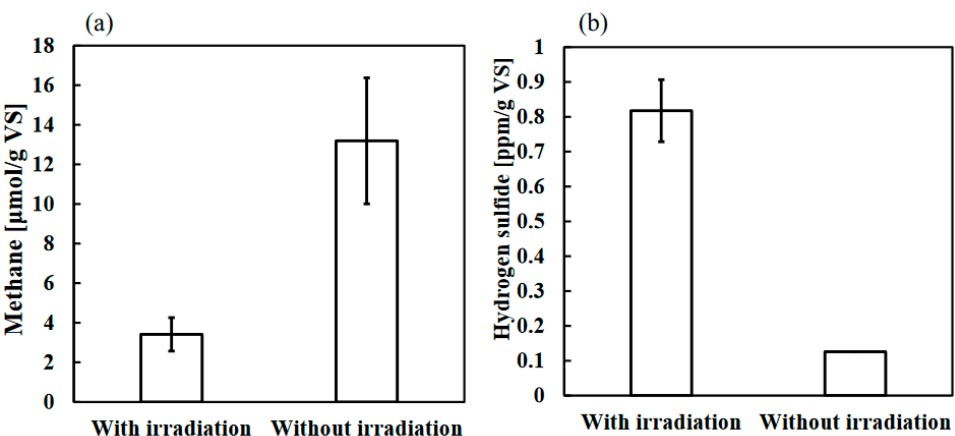

**Figure 5.** Effect of photo-irradiation on methanogenesis and sulfur-reducing activities: the activities of acetoclastic methanogens (**a**) and sulfate-reducing bacteria (**b**) were analyzed by measuring methane in acetate and hydrogen sulfide using the WSS samples with or without photo-irradiation for 7 days. Error bars represent standard errors (*n* = 3).

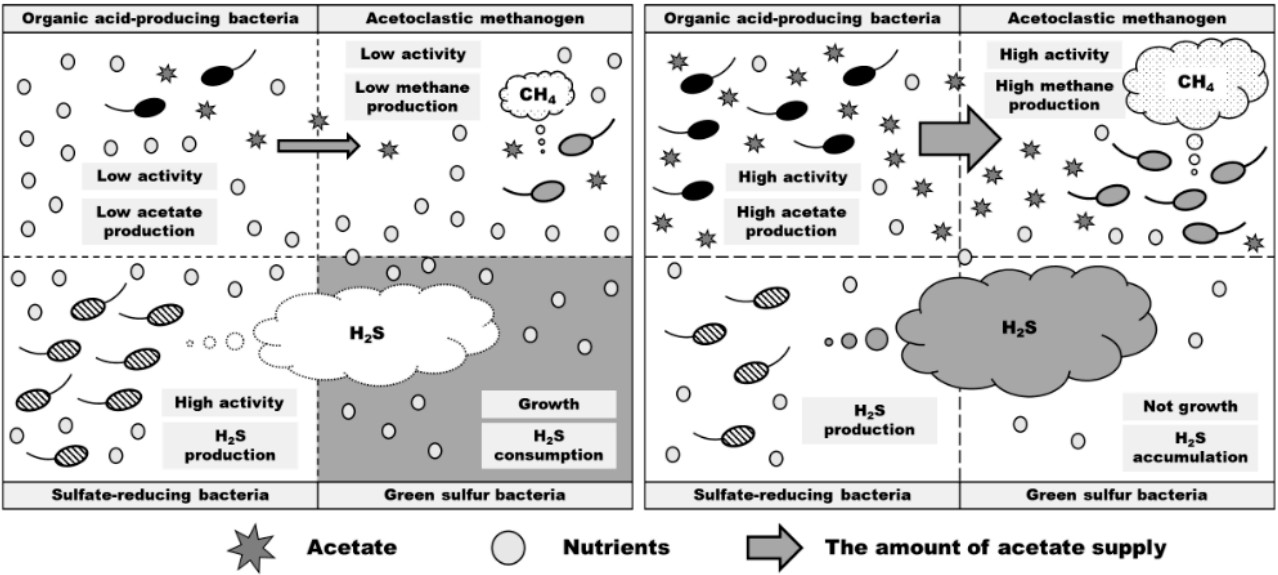

**Figure 6.** A suggested summary regarding the effect of photo-irradiation on the anaerobic digestion of WAS.

Since diverse microorganisms are present in WSS, the microorganisms are thought to establish direct or indirect relationships (such as mutualism, predation, parasitism, competition, commensalism, suppression, and neutralism) with each other [45]. Revealing the various bacterial interactions present in environmental samples is important because it provides new considerations: the effects and inter-relationships with respect to four microbial groups (organic-acid-producing bacteria, acetoclastic methanogen, sulfur-reducing bacteria, and green sulfur bacteria) via the photo-irradiation of the anaerobic digestion system. Firstly, it is noted that the purpose of these microorganisms is not to produce each metabolite (acetate, methane, and hydrogen sulfide) but to obtain energy using the metabolic pathways via which each metabolite is produced. As shown in Figure 4b, photo-irradiation significantly suppressed the accumulation of organic acids, and this suggests that photo-irradiation reduced the activity of organic-acid-producing bacteria. Therefore, it also caused a decrease in the concentration of acetate in the WSS (Figure 4b). Next, the reduced concentration of acetate in the WSS may have created an adverse environment for acetoclastic methanogens, which use acetate as a substrate to produce methane. Hence, methane production and acetoclastic methanogen activity in the photo-irradiation sample decreased (Figures 1a and 5a). In addition, the decrease in the activity of these microbial groups may have increased the relative amount of nutrients available to the sulfate-reducing bacteria in the WSS. Therefore, the activity of sulfur-reducing bacteria was high in the photo-irradiation sample (Figure 5b). However, the hydrogen sulfide produced by sulfur-reducing bacteria was consumed during the anoxygenic photosynthesis of green sulfur bacteria, and their growth was promoted via photo-irradiation. Thus, the amount of hydrogen sulfide was low in the photo-irradiation sample (Figure 1b).

### 3.4. Removal of Hydrogen Sulfide during Anaerobic Digestion via Dark/Light Switching

As mentioned above, the photo-irradiation of the anaerobic digestion system was shown to significantly reduce not only methane but also hydrogen sulfide. Since the presence of hydrogen sulfide during anaerobic digestion has a negative impact on methanogenesis efficiency [44], the removal of hydrogen sulfide is essential in long-term anaerobic digestion processes in order to protect against metal corrosion [46]. However, adversely affecting methanogenesis is undesirable because methane gas is a type of biogas [5]. Therefore, the amount of methane should be maintained but that of hydrogen sulfide can be removed. Hence, the practical usage of dark/light switching for anaerobic digestion using WSS was investigated. After 52 days of anaerobic digestion without photo-irradiation, the WSS samples were further incubated under light conditions. As a result, there was no reduction of hydrogen sulfide (the color of vials did not turn green). On the other hand, the supply of WSS with green sulfur bacteria and photo-irradiation was only able to reduce hydrogen sulfide while methane contents almost remained the same (Figure 7). Compared to the sample of WSS after 58 days of anaerobic digestion without photo-irradiation (control sample), no change in the amount of existing methane levels in the dark/light switch sample was observed after 6 days of incubation (Figure 7a). In addition, the amount of hydrogen sulfide at 58 days of incubation was not detected in the sample with respect to dark/light switching, even though the control sample contained $15 \pm 4$ ppm/g VS of hydrogen sulfide (Figure 7b). Thus, dark/light switching in long-term anaerobic digestion may only efficiently remove hydrogen sulfide without affecting the produced methane. However, continuous methane production during short-term anaerobic digestion was suppressed via photo-irradiation (Figure 1a). In addition, since hydrogen sulfide in anaerobic digestion causes the corrosion of the digester tank, its early removal is desirable [47]. Therefore, an anaerobic digestion switching system with respect to photo-irradiated and non-photo-irradiated WSS is proposed, and it should be further investigated in the future. Our previous research study reported efficient hydrogen sulfide removal in a contactless anaerobic digestion system using inorganic materials [27].

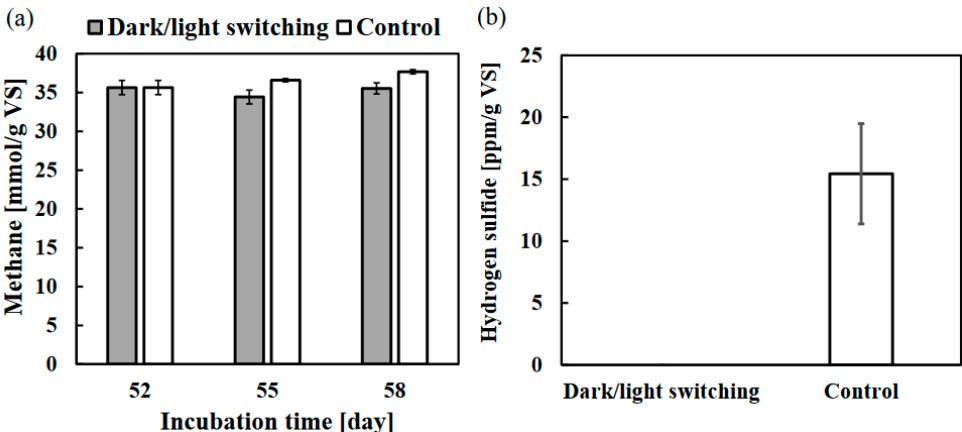

**Figure 7.** Effect of dark/light switching fermentation on the anaerobic digestion of WSS: methane (**a**) and hydrogen sulfide (**b**) were measured in vials that were originally incubated for 52 days without photo-irradiation and then further incubated for 6 days (a total of 58 days) with photo-irradiation after inoculating 0.5 mL of WSS with green sulfur bacteria. The amount of methane was measured at 52, 55, and 58 days of incubation. The amount of hydrogen sulfide was measured at the end of the incubation (58 days). Error bars represent standard errors (*n* = 3).

## 4. Conclusions

The effect of photo-irradiation on anaerobic digestion was investigated in comparison to the dark fermentation sample. The activity of organic-acid-producing bacteria in WSS was decreased via photo-irradiation, and this markedly suppressed the accumulation of organic acids in WSS (total concentration of organic acids: 0.4 mM and 3.1 mM with or without photo-irradiation). In addition, the activity of acetoclastic methanogens in the WSS was particularly reduced, and overall methane production was reduced 3.9-fold. In addition, it was suggested that the activity of sulfate-reducing bacteria in the WSS was not directly affected by photo-irradiation but indirectly increased up to 6.5-fold in terms of nutrient acquisition. Photo-irradiation relative to the anaerobic digestion system promoted the growth of green sulfur bacteria and caused the significant consumption of hydrogen sulfide via anoxygenic photosynthesis. Light/dark switching in the long-term anaerobic digestion system only efficiently removed hydrogen sulfide without affecting the produced methane gas. Therefore, this study not only reported the effects of photo-irradiation on microbial interactions within the anaerobic digestion system but also reported the potential of photo-irradiation as a method for efficient hydrogen sulfide removal. Taken together, the light/dark switching system produced 35 mmol/g VS of methane and undetectable hydrogen sulfide at 58 days, despite the presence of 37 mmol/g VS of methane and 15 ppm/g VS of hydrogen sulfide, without photo-irradiation.

**Author Contributions:** Conceptualization, T.M.; methodology, S.T., K.Y. and T.M.; investigation, S.T., K.Y. and S.I.; data curation, T.M.; writing—original draft preparation, S.T.; writing—review and editing, T.M.; supervision, T.M.; project administration, T.M.; funding acquisition, T.M. All authors have read and agreed to the published version of the manuscript.

**Funding:** This research was funded by the Research Foundation for Opto-Science and Technology.

**Institutional Review Board Statement:** Not applicable because this study did not involve humans or animals.

**Informed Consent Statement:** Not applicable because this study did not involve humans.

**Data Availability Statement:** All the data are available upon request.

**Acknowledgments:** We are grateful to the Research Foundation for Opto-Science and Technology for supporting this study.

**Conflicts of Interest:** The authors declare no conflict of interest.

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
