# Peer review of "Effect of Photo Irradiation on the Anaerobic Digestion of Waste Sewage Sludge-Reduced Methane and Hydrogen Sulfide Productions"

_fermentation, doi:10.3390/fermentation9110943_

Round 1

Reviewer 1 Report

Comments and Suggestions for Authors

Article entitled „Effect of Photo Irradiation on Anaerobic Digestion of Waste Sewage Sludge-Reduced Methane and Hydrogen Sulfide Productions” written by Shotaro Toya, Shunsuke Iriguchi , Kohei Yamaguchi , Toshinari Maeda and submitted to Fermentaion journal deals with  the effect of photo-fermentation (photo-irradiation) on anaerobic digestion using WSSs in in comparison to dark fermentation sample. The article is interesting and could be considered for publication in Fermentation journal. As English is not my native language, I am not able to assess language correctness. However, while reading, I conclude that the introduction, in which the information contained in it is often of a textbook nature. Knowledge on this topic is currently extensive and it certainly provides a great opportunity to enrich the content of the article by making references to the achievements and observations of other researchers in this area. Conclusions should be detailed, providing specific results of the research obtained.

Moreover, the quality of the figures should be adjusted to the requirements of the journal.

Author Response

Thank you very much for reviewing our manuscript.  We are glad to have the positive comments and important suggestions to improve our manuscript.

According to the  comments, we revised the manuscript appropriately.

1) We carefully explained some points of ideas so that readers can understand the research contents in the abstract, introduction, materials and methods, and results and discussion.

2) Also conclusion is also revised to state more specific information.

Therefore, the current version of our manuscript meets with the quality of the journal.

Reviewer 2 Report

Comments and Suggestions for Authors

The manuscript is interesting, novel and well written. The manuscript shows how photo-radiation affects the methane and H2S production during anaerobic digestion. My minor comments are:

1. In the abstract, the authors must present the results in values that demonstrate the impact of the operating conditions during anaerobic digestion.

2. Authors should present methane production in terms of specific methane activity, methane yield or cumulative methane. Apply the modified Gompertz model.

3. Figure 1a, for with irradiation test, the error bars are not observed.

Author Response

Thank you very much for reviewing our manuscript and we are glad to hear of the positive comments which are indeed useful to improve our manuscript better.

  1. We revised abstract to state more specific information and number.
  2. The modified Gompertz model is a model simulation to estimate the cumulative methane. Our study is a fundamental research; therefore, we measured methane in a closed system, not open/sequencial system. .  The data do not need to be simulated by using the model.  Our data showing the amount of methane is a real amount; so, the current disply of methane production is more accurate rather than using such a model.  By the way, this model should be used in the future study (in more industrial situation). So the authors may study on the model so that we can use it in the furure.
  3. Actually, the error bars are small so they are not viable. So we revised the figure so that all the readers can see the errror bars in Fig. 1a.